# A Model for the Generalised Dispersion of Synovial Fluids on Nutritional Transport with Joint Impacts of Electric and Magnetic Field

B. Rushi Kumar [1,*], R. Vijayakumar [2,3] and A. Jancy Rani [4]

1   Department of Mathematics, School of Advanced Sciences, Vellore Institute of Technology,
    Vellore 632014, Tamil Nadu, India
2   Mathematics Section, Faculty of Engineering and Technology, Annamalai University, Annamalai Nagar,
    Chidambaram 608002, Tamil Nadu, India
3   Department of Mathematics, Periyar Arts College, Cuddalore 607001, Tamil Nadu, India
4   Department of Mathematics, Annamalai University, Annamalai Nagar,
    Chidambaram 608002, Tamil Nadu, India
*   Correspondence: rushikumar@vit.ac.in

**Abstract:** This work analyses the effect of electromagnetic fields on cartilaginous cells in human joints and the nutrients that flow from the synovial fluid to the cartilage. The perturbation approach and the generalised dispersion model is used to solve the governing equation of momentum and mass transfer. The dispersion coefficient increases with dimensionless time. It aids in grasping the level of nutritional transport to the synovial joint. Low-molecular-weight solutes have a lower concentration distribution at the same depth in articular cartilage than high-molecular-weight solutes. Thus, diffusion dominates nutrition transport for low-molecular-weight solutes, whereas a mechanical pumping action dominates nutrition transport for high-molecular-weight solutes. The report says that the cells in the centre of the cartilage surface receive more nutrients during imbibition and exudation than the cells on the periphery, and the earliest indications of cartilage degradation emerge in the uninflected regions. As a result, cartilage nutrition is considered necessary to joint mobility. It also predicts that, as the viscoelastic parameter increases, the concentration in the articular cartilage diminishes, resulting in the cartilage cells receiving less nutrition, which might lead to harmful effects. The dispersion coefficient and mean concentration for distinct factors, such as the Hartmann number, porous parameter, and viscoelastic parameters of gel formation, have been computed and illustrated through graphics.

**Keywords:** synovial fluid; electromagnetic fields; generalized dispersion model

**MSC:** 76D05; 76S05; 76W05

## 1. Introduction

The synovial joint, which is responsible for the biomechanics of the knee joint, plays an essential role in the movement of living things such as humans and animals. A load-bearing bone with protected ends makes up a synovial joint. Knee joints are one of the body's most extensive and complicated joints. According to Alshehri and Sharma [1], articular cartilage is the thin layer of connective tissue that protects the articulating ends of bones in synovial joints (movable joints in the body, such as the knee, hip, and shoulder).

The primary idea of this study is how magnetotherapy aids in the treatment of human joint cartilage, rheumatoid illnesses, and other diseases, such as osteoporosis. Articular cartilage and synovial fluid work together to keep joints lubricated. Damage to the articular cartilage may cause the synovial fluid to have poor rheological qualities, eventually affecting the joints' performance. Due to their importance in joint lubrication, the rheological features of synovial fluid are of interest. The arrangement and functions of human

joints are covered in synovial anatomy and physiology. Flow mechanics, heat and mass transfer, and reaction kinetics influence synovial bio-lubrication. According to Zahn and Shenton [2], both electric and magnetic fields affect current conductivity through tissue (cartilage) and lubricant. The electric field inside the body reflects the movement of current through the conducting body tissues and affects the system's physiological behaviors. A magnetic field surrounds the body. This would induce currents to travel in circuitous directions if it were clear.

Researchers are interested in studying non-Newtonian and MHD flow because of their various physical configurations and applications. Akbar et al. [3] explored the peristaltic flow of Jeffrey nanofluid convective boundary constraints in an asymmetric channel. Faghiri et al. [4] examined the non-Newtonian fluid in a circular tube with a non-uniform heat flux. An analysis of the hydromagnetic hyperbolic-tangent liquid with radiation, a heat source, and stratification was determined by Gulzar et al. [5]. Based on a convective boundary condition, Khan et al. [6] numerically delineated the magnetic field on the boundary layer of Sisko liquid at the surface. Attar et al. [7] introduced an analytical solution to nonlinear fractional differential equations using Akbari-Ganji's method. Hossain et al. [8] depicted the effects of a cylinder on natural convection in a square cavity. Bhuvaneswari et al. [9] examined the magnetoconvection within a cavity with a magnetic effect. Sivasankaran et al. [10] investigated the MHD mixed convection in a lid-driven cavity. The effect of MHD on microchannel heat sinks was deliberated by Narrein et al. [11]. Sivasankaran and Narrein [12] studied the MHD convective flow in a trapezoidal microchannel heat sink. Sivasankaran et al. [13] also presented the MHD discrete heating in free convection in a porous container. Several researchers have completed significant work on hydromagnetic convective effects in recent years, including Sivasankaran et al. [14]; Bindhu et al. [15]; Bhuvaneswari et al. [16]; Rashad et al. [17]; Niranjan et al. [18]. Other related articles have also appeared in recent publications [19–25].

Tandon et al. [26] have consistently proposed the use of magnetic fields for treating synovial joints. According to Rudraiah et al. [27], finding the dispersion coefficient using Gill and Sankarasubramanian's [28] generalised dispersion model is beneficial in the dispersion of nutrients in synovial fluid. Using Taylor's dispersion model, Ng et al. [29] examined the effect of a fixed charge density on the electrohydrodynamic transport of synovial fluid constituents such as hyaluronic acid, glycoprotein, and other molecules to artificial or natural joints. Khan et al. [30] investigated the effect of the generated magnetic field on synovial fluid in an asymmetric channel with the peristaltic movement of non-Newtonian fluid. Nagaraj et al. [31] used the Taylor dispersion coefficient to investigate nutrients' dispersion of synovial fluid into cartilage under the effect of electric and magnetic fields for both artificial and natural joints. There are some other related papers in recent publications [32,33].

In a flow of synovial fluid, Ramakrishnan and Swetha [34] investigated the thickness on the porous plate on axial velocity and skin friction in the human joints with teh required BJR slip states. Beretta et al. [35] reviewed the scientific literature on the effect of applied fields on microorganisms. Vijayakumar and Ratchagar [36] conducted a thorough analysis of unsteady convective diffusion to look at how nutrients and other proteins are transported from synovial fluid to articular cartilage. The authors of this paper investigated the model for the synovial fluid, also known as joint fluid and located in the knee joints, by examining the nutritional transportation of generalised dispersion with the effect of an electric and magnetic field. The behaviour of the synovial fluid was analysed using the perturbation technique and the generalised dispersion model. The exact solution is plotted graphically and explained in detail.

## 2. Formulation of the Problem

We introduced the following acceptable assumptions to define a mathematically tractable problem. Knee joints are among the body's most complex and essential joints. The knee connects the femur (thighbone) to the shinbone (tibia). Figure 1 represents the physical

configuration of the human knee joint (Tandon et al. [37], and Alshehri and Sharma [1]). Viscoelastic fluid was used to represent synovial fluid because of its elasticity, which is important for lubricating joints. Articular cartilage is a very elastic material. Introducing the standard lubrication theory hypothesis into the Navier–Stokes equation of motion and ignoring the variation in pressure normal to very thin lubrication films, the investigation is subjective, with the following assumptions:

- A 2D, electrically conducting, viscous and incompressible synovial fluid is considered.
- Flow of fluid is laminar and steady.
- A constant magnetic field of strength $B_0$ is applied in the transverse direction.

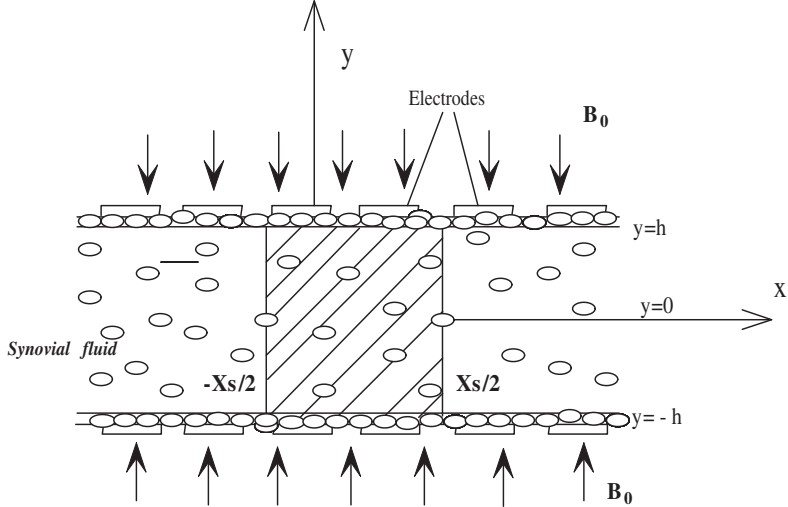

**Figure 1.** Physical Configuration of the human knee joint.

Under these assumptions, the governing differential equations of momentum, continuity, and concentration of the fluid film (synovial fluid) region are as follows: [37,38]:

$$0 = -\frac{\partial \hat{p}}{\partial \hat{x}} + \hat{\eta}\frac{\partial}{\partial \hat{y}}\left[\frac{\partial \hat{u}}{\partial \hat{y}} - K_0\left(\frac{\partial \hat{u}}{\partial \hat{y}}\right)^3\right] - B_0^2\sigma_0\hat{u} - \frac{\mu}{k}\hat{u} + \hat{\rho}_e\hat{E}_x \tag{1}$$

$$\frac{\partial \hat{u}}{\partial \hat{x}} + \frac{\partial \hat{v}}{\partial \hat{y}} = 0 \tag{2}$$

$$\frac{\partial \hat{C}}{\partial \hat{t}} + \hat{u}\frac{\partial \hat{C}}{\partial \hat{x}} = \hat{D}\left(\frac{\partial^2 \hat{C}}{\partial \hat{x}^2} + \frac{\partial^2 \hat{C}}{\partial \hat{y}^2}\right) \tag{3}$$

with boundary conditions

$$\frac{\partial \hat{u}}{\partial \hat{y}} = \frac{-\alpha}{\sqrt{k}}\hat{u} \quad at \quad \hat{y} = h, \tag{4}$$

$$\frac{\partial \hat{u}}{\partial \hat{y}} = \frac{\alpha}{\sqrt{k}}\hat{u} \quad at \quad \hat{y} = -h, \tag{5}$$

$$\hat{C}(0, \hat{x}, \hat{y}) = \begin{cases} \hat{C}_0, |\hat{x}| \leq \dfrac{\hat{x}_s}{2} \\ 0, |\hat{x}| > \dfrac{\hat{x}_s}{2} \end{cases} \tag{6}$$

$$\frac{\partial \hat{C}}{\partial \hat{y}}(\hat{t}, \hat{x}, -h) = \frac{\partial \hat{C}}{\partial \hat{y}}(\hat{t}, \hat{x}, h) = 0 \tag{7}$$

$$\hat{C}(\hat{t}, \infty, \hat{y}) = \frac{\partial \hat{C}}{\partial \hat{x}}(\hat{t}, \infty, \hat{y}) = 0 \tag{8}$$

Equations (4) and (5) represents Beavers and Joseph [39] slip condition at lower and higher porous surfaces.

The dimensionless form of Equations (1) and (3) is created using

$$u = \frac{\hat{u}}{\tilde{u}}; \quad \xi_s = \frac{\hat{x}_s}{hPe}; \quad \eta = \frac{\hat{y}}{h}; \quad p = \frac{\hat{p}}{\rho \tilde{u}^2}; \quad \xi' = \frac{\hat{x}}{hPe}; \quad \tau = \frac{\hat{D}\hat{t}}{h^2}; \quad \rho_e = \frac{\hat{\rho}_e h}{\epsilon_0 V}; \quad E_x = \frac{\hat{E}_x h}{V};$$

$$\phi = \frac{\hat{C}}{\hat{C}_0};$$

Hence,

$$\frac{\partial^2 u}{\partial \eta^2} - 3 \epsilon \left(\frac{\partial u}{\partial \eta}\right)^2 \frac{\partial^2 u}{\partial \eta^2} - s_2 u = s_3(1 - \alpha_c \eta) + s_1 \tag{9}$$

and

$$\frac{\partial \phi}{\partial \tau} + u \frac{\partial \phi}{\partial \xi'} = \frac{1}{Pe^2} \frac{\partial^2 \phi}{\partial \xi'^2} + \frac{\partial^2 \phi}{\partial \eta^2} \tag{10}$$

where, $s_1 = \frac{Re}{Pe} \frac{\partial p}{\partial \xi}$, $s_2 = M^2 + \frac{1}{Da}$, $s_3 = \frac{WePeX_0\alpha_c}{2}$, $M^2 = \frac{B_0^2 \sigma_0 h^2}{\hat{\eta}}$, $We = \frac{\epsilon \tilde{u}^2}{\hat{\eta}}$, $Re = \frac{\rho \tilde{u} h}{\hat{\eta}}$, $Pe = \frac{\tilde{u}h}{\hat{D}}$, $\sigma = \frac{h}{\sqrt{k}}$, $\epsilon = \frac{K_0 \tilde{u}^2}{h^2}$, $Da = \frac{k}{h^2}$, $\hat{\eta} = \frac{\mu}{\rho}$

Axial coordinate moving with an average velocity is defined as $\hat{x}_1 = \hat{x} - t\tilde{u}$ and its non-dimensional form $\xi = \xi' - \tau$, $\xi = \frac{\hat{x}_1}{hPe}$. Now, (10) becomes,

$$\frac{\partial \phi}{\partial \tau} + \breve{U} \frac{\partial \phi}{\partial \xi} = \frac{1}{Pe^2} \frac{\partial^2 \phi}{\partial \xi^2} + \frac{\partial^2 \phi}{\partial \eta^2} \tag{11}$$

with conditions of $\psi$

$$\frac{\partial u}{\partial \eta} = -\alpha \sigma u \quad at \quad \eta = 1 \tag{12}$$

$$\frac{\partial u}{\partial \eta} = \alpha \sigma u \quad at \quad \eta = -1 \tag{13}$$

$$\phi(0, \xi, \eta) = \begin{cases} 1, |\xi| \leq \frac{\xi_s}{2} \\ 0, |\xi| > \frac{\xi_s}{2} \end{cases} \tag{14}$$

$$\frac{\partial \phi}{\partial \eta}(\tau, \xi, -1) = \frac{\partial \phi}{\partial \eta}(\tau, \xi, 1) = 0 \tag{15}$$

$$\phi(\tau, \infty, \eta) = \frac{\partial \phi}{\partial \xi}(\tau, \infty, \eta) = 0 \tag{16}$$

## 3. Method of Solution

### 3.1. Velocity Distribution

Let us incorporate the perturbation approach to solve (9) as follows:

$$u = u_0 + \epsilon \, u_1 + O(\epsilon^2) \tag{17}$$

where,

$$u_0 = A_1 e^{-\sqrt{s_2}\eta} + A_2 e^{\sqrt{s_2}\eta} - \frac{s_1}{s_2} - \frac{s_3}{s_2^2}(1 - \alpha_c \eta) \tag{18}$$

$$u_1 = 2C_3 \cosh M\eta - \frac{3}{2}C_1^3 M^3 \left( \frac{\cosh 3M\eta}{4M} - \eta \sinh 3M\eta \right) \tag{19}$$

$$s_4 = (\alpha\sigma - \sqrt{s_2}), \ s_5 = (\alpha\sigma + \sqrt{s_2}), \ s_6 = -\frac{s_3}{s_2^2}\alpha_c + \alpha\sigma\left(\frac{s_1}{s_2} + \frac{s_3}{s_2^2}\right), \ s_7 = s_5 e^{-\sqrt{s_2}},$$

$$s_8 = s_4 e^{\sqrt{s_2}}, \ s_9 = -\left[\frac{s_3}{s_2^2}\alpha_c + \alpha\sigma\left(\frac{s_1}{s_2} + \frac{s_3}{s_2^2}(1 - \alpha_c)\right)\right],$$

$$A_1 = \frac{1}{s_4}(s_6 - s_5 A_2), \ A_2 = \frac{s_9(s_6 - s_4)}{(s_7 s_5 - s_4 s_9)},$$

The normalized axial velocity components acquired from Equation (17) are

$$\breve{U} = \frac{u - \tilde{u}}{\tilde{u}}$$

where,

$$\tilde{u} = 2\left(\frac{A_1(1 - e^{-\sqrt{s_2}}) + A_2(-1 + e^{\sqrt{s_2}})}{\sqrt{s_2}}\right) + 2s_{20} + \alpha_c +$$

$$\frac{(1 - e^{-3\sqrt{s_2}})s_{10}\epsilon + (-1 + e^{3\sqrt{s_2}})s_{11}\epsilon}{12 s_2^{\frac{3}{2}}} + \frac{(1 - e^{-2\sqrt{s_2}})s_{14}\epsilon + (-1 + e^{2\sqrt{s_2}})s_{15}\epsilon}{3 s_2^{\frac{3}{2}}} +$$

$$\frac{(1 + e^{\sqrt{s_2}}(-1 + \sqrt{s_2}))s_{12}\epsilon + (1 - e^{-\sqrt{s_2}}(1 + \sqrt{s_2}))s_{13}\epsilon}{s_2^{\frac{3}{2}}} +$$

$$\frac{A_3(1 - e^{-\sqrt{s_2}})\epsilon + A_4(-1 + e^{\sqrt{s_2}})\epsilon}{\sqrt{s_2}}, \ s_{10} = 2\epsilon A_1^2 s_2^2, \ s_{11} = 2\epsilon A_2^2 s_2^2,$$

$$s_{12} = \frac{A_2 s_3^2 \alpha_c}{s_2^2} - A_1 A_2^2 s_2^2, \ s_{13} = \frac{A_1 s_3^2 \alpha_c}{s_2^2} - A_1^2 A_2 s_2^2, \ s_{14} = \frac{2A_1^2 s_3 \alpha_c}{\sqrt{s_2}}, \ s_{15} = \frac{2A_2^2 s_3 \alpha_c}{\sqrt{s_2}},$$

$$s_{16} = -\left(\frac{3(-s_{10} + s_{11})}{8\sqrt{s_2}} + \frac{s_{12} + s_{13}}{2\sqrt{s_2}} - \frac{2(s_{14} - s_{15})}{3\sqrt{s_2}} + \alpha\sigma\left(\frac{s_{10} + s_{11}}{8s_2} + \frac{s_{14} + s_{15}}{3s_2}\right)\right),$$

$$s_{17} = s_5 e^{-\sqrt{s_2}}, \ s_{18} = s_4 e^{\sqrt{s_2}}, \ s_{20} = \frac{s_1}{s_2} - \frac{s_3}{s_2^2}, \ s_{19} =$$

$$-\frac{3(-s_{10}e^{-3\sqrt{s_2}} + s_{11}e^{3\sqrt{s_2}})}{8\sqrt{s_2}} + \frac{s_{12}e^{\sqrt{s_2}} + s_{13}e^{-\sqrt{s_2}}}{2\sqrt{s_2}} + \frac{2(s_{14}e^{-2\sqrt{s_2}} - s_{15}e^{2\sqrt{s_2}})}{3\sqrt{s_2}} +$$

$$\alpha\sigma\left(\frac{s_{10}e^{-3\sqrt{s_2}} + s_{11}e^{3\sqrt{s_2}}}{8\sqrt{s_2}} \frac{s_{12}e^{\sqrt{s_2}} + s_{13}e^{-\sqrt{s_2}}}{2\sqrt{s_2}} + \frac{s_{14}e^{-2\sqrt{s_2}} - s_{15}e^{2\sqrt{s_2}}}{3\sqrt{s_2}}\right)$$

### 3.2. Generalized Dispersion Model (GDM)

In order to establish the mean concentration that is valid for $\tau$, we adopted the GDM of Gill and Sankarasubramanian [28].

$$\phi(\tau, \xi, \eta) = \phi_m(\tau, \xi) + \sum_{k=1}^{\infty} f_k(\tau, \eta)\frac{\partial^k \phi_m}{\partial \xi^k} \tag{20}$$

where,

$$\phi_m(\tau, \xi) = \frac{1}{2}\int_{-1}^{1} \phi(\tau, \xi, \eta)d\eta \tag{21}$$

Integrating (11), we obtain

$$\frac{\partial \phi_m}{\partial \tau} = \frac{1}{Pe^2}\frac{\partial^2 \phi_m}{\partial \xi^2} + \frac{1}{2}\int_{-1}^{1}\frac{\partial^2 \phi}{\partial \eta^2}\,d\eta - \frac{1}{2}\frac{\partial}{\partial \xi}\int_{-1}^{1}\breve{U}\,\phi\,d\eta \tag{22}$$

Using Equation (20) in (22), we obtain

$$\frac{\partial \phi_m}{\partial \tau} = \frac{1}{P_e^2}\frac{\partial^2 \phi_m}{\partial \xi^2} - \frac{1}{2}\frac{\partial}{\partial \xi}\int_{-1}^{1} \breve{U}\left(\phi_m(\tau,\xi) + f_1(\tau,\eta)\frac{\partial \phi_m}{\partial \xi}(\tau,\xi) + \dots\right)d\eta \tag{23}$$

Rewriting the above equation, we obtain

$$\frac{\partial \phi_m}{\partial \tau} = \sum_{k=1}^{\infty} K_k(\tau)\frac{\partial^k \phi_m}{\partial^k \xi} \tag{24}$$

Making use of Equation (24) in (23) we obtain

$$K_1\frac{\partial \phi_m}{\partial \xi} + K_2\frac{\partial^2 \phi_m}{\partial \xi^2} + \quad K_3\frac{\partial^3 \phi_m}{\partial \xi^3} + \dots = \frac{1}{P_e^2}\frac{\partial^2 \phi_m}{\partial \xi^2} - \frac{1}{2}\frac{\partial}{\partial \xi}\int_{-1}^{1} \breve{U}(\phi_m(\tau,\xi)$$

$$+ f_1(\tau,\eta)\frac{\partial \phi_m}{\partial \xi} + f_2(\tau,\eta)\frac{\partial^2 \phi_m}{\partial \xi^2}(\tau,\xi) + \dots)d\eta \tag{25}$$

Comparing the coefficient $\frac{\partial \phi_m}{\partial \xi}, \frac{\partial^2 \phi_m}{\partial \xi^2} \dots$ we get,

$$K_i(\tau) = \frac{\delta_{ij}}{P_e^2} - \frac{1}{2}\int_{-1}^{1} U f_{i-1}(\tau,\eta)d\eta, \quad (i = 1,2,3,\dots \ and \ j = 2) \tag{26}$$

where, Kroneckar delta $\delta_{ij} = \begin{cases} 1, & if \quad i = j \\ 0, & if \quad i \neq j \end{cases}$

Incorporating Equation (20) in (11), we acquire

$$\frac{\partial}{\partial \tau}\left(\phi_m(\tau,\xi) + f_1(\tau,\eta)\frac{\partial \phi_m}{\partial \xi}(\tau,\xi) + f_2(\tau,\eta)\frac{\partial^2 \phi_m}{\partial \xi^2}(\tau,\xi) + \dots\right)$$

$$+ \breve{U}\frac{\partial}{\partial \xi}\left(\phi_m(\tau,\xi) + f_1(\tau,\eta)\frac{\partial \phi_m}{\partial \xi}(\tau,\xi) + f_2(\tau,\eta)\frac{\partial^2 \phi_m}{\partial \xi^2}(\tau,\xi) + \dots\right)$$

$$= \frac{1}{P_e^2}\frac{\partial^2}{\partial \xi^2}\left(\phi_m(\tau,\xi) + f_1(\tau,\eta)\frac{\partial \phi_m}{\partial \xi}(\tau,\xi) + f_2(\tau,\eta) + \dots\right)$$

$$+ \frac{\partial^2}{\partial \eta^2}\left(\phi_m(\tau,\xi) + f_1(\tau,\eta)\frac{\partial \phi_m}{\partial \xi} + \dots\right) \tag{27}$$

Modifying the terms and employing

$$\frac{\partial^{k+1} \phi_m}{\partial \tau \partial \xi^k} = \sum_{i=1}^{\infty} K_i(\tau)\frac{\partial^{k+i} \phi_m}{\partial \xi^{k+i}}$$

we obtain

$$\left[\frac{\partial f_1}{\partial \tau} - \frac{\partial^2 f_1}{\partial \eta^2} + \breve{U} + K_1(\tau)\right]\frac{\partial \phi_m}{\partial \xi} + \left[\frac{\partial f_2}{\partial \tau} - \frac{\partial^2 f_2}{\partial \eta^2} + f_1\breve{U} + K_1(\tau)f_1 + K_2(\tau) - \frac{1}{P_e^2}\right]\frac{\partial^2 \phi_m}{\partial \xi^2}$$

$$+ \sum_{k=1}^{\infty}\left[\frac{\partial f_{k+2}}{\partial \tau} - \frac{\partial^2 f_{k+2}}{\partial \eta^2} + f_{k+1}\breve{U} + f_{k+1}K_1(\tau) + \left(K_2(\tau) - \frac{1}{P_e^2}\right)f_k\right.$$

$$\left. + \sum_{i=3}^{k+2} K_i f_{k+2-i}\right]\frac{\partial^{k+2} \phi_m}{\partial \xi^{k+2}} = 0 \tag{28}$$

with $f_0 = 1$. Comparing the coefficients of $\frac{\partial^k \phi_m}{\partial \xi^k}$ $(k = 1, 2, 3, \ldots)$ in (28) and setting it equal to zero, we obtain

$$\frac{\partial f_1}{\partial \tau} = \frac{\partial^2 f_1}{\partial \eta^2} - \breve{U} - K_1(\tau) \tag{29}$$

$$\frac{\partial f_2}{\partial \tau} = \frac{\partial^2 f_2}{\partial \eta^2} - f_1 \breve{U} - K_1(\tau) f_1 - K_2(\tau) + \frac{1}{P_e^2} \tag{30}$$

$$\frac{\partial f_{k+2}}{\partial \tau} = \frac{\partial^2 f_{k+2}}{\partial \eta^2} - f_{k+1} \breve{U} - K_1(\tau) f_{k+1} - \left( K_2(\tau) - \frac{1}{P_e^2} \right) f_k - \sum_{i=3}^{k+2} K_i f_{k+2-i} \tag{31}$$

To obtain $K_i$'s we have $f_k$'s and its respective initial and boundary constraints,

$$f_k(0, \eta) = 0 \tag{32}$$

$$\frac{\partial f_k}{\partial \eta}(\tau, -1) = 0 \tag{33}$$

$$\frac{\partial f_k}{\partial \eta}(\tau, 1) = 0 \tag{34}$$

$$\int_{-1}^{1} f_k(\tau, \eta) d\eta = 0, \tag{35}$$

for $k = 1, 2, 3, \ldots$

From Equation (26) for $i = 1$, we obtain

$$K_1(\tau) = 0 \tag{36}$$

From Equation (26) for $i = 2$, we obtain $K_2$ as,

$$K_2(\tau) = \frac{1}{P_e^2} - \frac{1}{2} \int_{-1}^{1} \breve{U} f_1 d\eta \tag{37}$$

To evaluate $K_2(\tau)$,

$$\text{let } f_1 = f_{10}(\eta) + f_{11}(\tau, \eta) \tag{38}$$

where $f_{10}(\eta)$ is not dependent on $\tau$ and pertains to an infinite wide slug and $f_{11}$ is $\tau$-dependent, satisfying

$$\frac{d f_{10}}{d\eta} = 0 \text{ at } \eta = \pm 1 \tag{39}$$

$$\int_{-1}^{1} f_{10} d\eta = 0 \tag{40}$$

Using the (38) in (29) gives

$$\frac{d^2 f_{10}}{d\eta^2} = \breve{U} \tag{41}$$

$$\frac{\partial f_{11}}{\partial \tau} = \frac{\partial^2 f_{11}}{\partial \eta^2} \tag{42}$$

Solving the Equation (41) with conditions (39) and (40) is

$$f_{10} = \frac{1}{72s_2^2} e^{-3\sqrt{s_2}\eta} (-s_{10}\epsilon - e^{6\sqrt{s_2}\eta}s_{11}\epsilon - 6e^{\sqrt{s_2}\eta}s_{14}\epsilon - 6e^{5\sqrt{s_2}\eta}s_{15}\epsilon +$$

$$36e^{4\sqrt{s_2}\eta}(2A_2s_2 - \epsilon(2A_4s_2 + s_{12}(-2 + \sqrt{s_2}\eta))) +$$

$$36e^{2\sqrt{s_2}\eta}(2A_1s_2 - \epsilon(2A_3s_2 + s_{13}(2 + \sqrt{s_2}\eta))) +$$

$$12e^{3\sqrt{s_2}\eta}s_2^2(3s_{20}\eta^2 + \alpha_c\eta^3 + 6(A_5 + \eta A_6))) \tag{43}$$

where

$A_5 = -\frac{1}{48\sqrt{s_2}s_2 e^{3\sqrt{s_2}}}(12e^{3\sqrt{s_2}}s_2(-2A_1 + 2A_2 + 2\epsilon(A_3 - A_4) + \sqrt{s_2}(\alpha c + 2s_{20})) -$

$12e^{2\sqrt{s_2}}(2A_1s_2 - \epsilon(2A_3s_2 + s_{13}\sqrt{s_2} + s_{13})) +$

$12e^{4\sqrt{s_2}}(2A_2s_2 + \epsilon(-2A_4s_2 + s_{12}(-\sqrt{s_2}) + s_{12})) + e^{3\sqrt{s_2}}(\epsilon(s_{10} - s_{11} + 4(3s_{12} + 3s_{13} + s_{14} - s_{15}))) +$

$s_{10}\epsilon - s_{11}e^{6\sqrt{s_2}}\epsilon + 4s_{14}e^{\sqrt{s_2}}\epsilon - 4s_{15}e^{5\sqrt{s_2}}\epsilon);$

$A_6 = -\frac{1}{9\,12s_2^{3/2}s_2 e^{3\sqrt{s_2}}}(9e^{3\sqrt{s_2}}s_2(24s_{A_1} - 24s_{A_2} + 24\epsilon(s_{A_4} - s_{A_3})) -$

$108e^{2\sqrt{s_2}}(2s_{A_1}s_2 - \epsilon(2s_{A_3}s_2 + s13(\sqrt{s_2} + 3))) +$

$108e^{4\sqrt{s_2}}(2s_{A_2}s_2 - \epsilon(2s_{A_4}s_2 + s_{12}(\sqrt{s_2} - 3))) + 9e^{3\sqrt{s_2}}s_2^{3/2}s_2(24s_{A_5} + \alpha c + 4s_{20}) +$

$e^{3\sqrt{s_2}}\epsilon(-(s_{10} - s_{s_{11}} + 9(36s_{12} + 36s13 + s_{14} - s_{15}))) + s_{10}\epsilon - s_{s_{11}}e^{6\sqrt{s_2}}\epsilon + 9s_{14}e^{\sqrt{s_2}}\epsilon - 9s_{15}e^{5\sqrt{s_2}}\epsilon)$

Equation (42) represents thermal conduction and its solution satisfies $f_{11}(\tau, \eta) = -f_{10}(\eta)$

$$f_{11} = \sum_{n=1}^{\infty} B_n e^{-\lambda_n^2\tau} \cos(\lambda_n\eta) \tag{44}$$

$$\text{where,} \quad B_n = -2\int_0^1 f_{10}(\eta)\cos(\lambda_n\eta)d\eta$$

Substituting (43) and (44) in Equation (38), we obtain,

$f_1 = \frac{1}{72s_2^2} e^{-3\sqrt{s_2}\eta}(-s_{10}\epsilon - e^{6\sqrt{s_2}\eta}s_{11}\epsilon - 6e^{\sqrt{s_2}\eta}s_{14}\epsilon - 6e^{5\sqrt{s_2}\eta}s_{15}\epsilon + 36e^{4\sqrt{s_2}\eta}(2A_2s_2 -$

$\epsilon(2A_4s_2 + s_{12}(-2 + \sqrt{s_2}\eta))) + 36e^{2\sqrt{s_2}\eta}(2A_1s_2 - \epsilon(2A_3s_2 + s_{13}(2 + \sqrt{s_2}\eta))) +$

$12e^{3\sqrt{s_2}\eta}s_2^2(3s_{20}\eta^2 + \alpha_c\eta^3 + 6(A_5 + \eta A_6)) + \frac{e^{\alpha_c^2\tau - 3\sqrt{s_2} - \pi^2\tau}\cos(\pi\eta)}{24\pi^4 s_2^{\frac{3}{2}}(s_2 + \pi^2)^2(4s_2 + \pi^2)(9s_2 + \pi^2)}$

$(\pi^4 s_{10}(\pi^2 + s_2)^2(\pi^2 + 4s_2)\epsilon - e^{6\sqrt{s_2}}\pi^4 s_{11}(\pi^2 + s_2)^2(\pi^2 + 4s_2)\epsilon + 4e^{\sqrt{s_2}}\pi^4 s_{14}(\pi^2 + s_2)^2(\pi^2 + 9s_2)\epsilon -$

$4e^{5\sqrt{s_2}}\pi^4 s_{15}(\pi^2 + s_2)^2(\pi^2 + 9s_2)\epsilon - 12e^{2\sqrt{s_2}}\pi^4(\pi^2 + 4s_2)(\pi^2 + 9s_2)(2A_1s_2(\pi^2 + s_2) -$

$(s_2(s_{13}(3 + \sqrt{s_2} + 2A_3s_2) + \pi^2(s_{13} + s_{13}\sqrt{s_2} + 2A_3s_2))\epsilon) + 12e^{4\sqrt{s_2}}\pi^4(\pi^2 + 4s_2)(\pi^2 + 9s_2)(2A_2s_2(\pi^2$

$+ s_2) - (s_2(s_{12}(-3 + \sqrt{s_2} + 2A_4s_2) + \pi^2(s_{12}(-1 + \sqrt{s_2}) + 2A_4s_2))\epsilon) + e^{3\sqrt{s_2}}(-24A_1\pi^4 s_2(\pi^2 + s_2)(\pi^2$

$+ 4s_2)(\pi^2 + 9s_2) + 24A_2\pi^4 s_2(\pi^2 + s_2)(\pi^2 + 4s_2)(\pi^2 + 9s_2) + 12s_2^{\frac{3}{2}}(\pi^2 + s_2)^2(\pi^2 + 4s_2)(\pi^2 +$

$9s_2)(-4\alpha_c + \pi^2(4A_6 + 2s_{20} + \alpha_c)) + \pi^4(2\pi^4 s_2(3s_{10} - 3s_{11} + 96s_{12} + 96s_{13} + 22s_{14} - 22s_{15} +$

$168(A_3 - A_4)s_2) + \pi^2 s_2^2(9s_{10} - 9s_{11} + 900s_{12} + 900s_{13} + 76s_{14} - 76s_{15} + 1176(A_3 - A_4)s_2) +$

$4s_2^3(s_{10} - s_{11} + 9(36s_{12} + 36s_{13} + s_{14} - s_{15} + 24(A_3 - A_4)s_2) + \pi^6(s_{10} - s_{11} + 4(3s_{12} + 3s_{13} +$

$s_{14} - s_{15} + 6A_3s_2 - 6A_4s_2)))\epsilon)) + \frac{e^{\alpha_c^2\tau - 3\sqrt{s_2} - 4\pi^2\tau}\cos(2\pi\eta)}{24\pi^2 s_2^{\frac{3}{2}}(s_2 + \pi^2)(s_2 + 4\pi^2)^2(9s_2 + 4\pi^2)}(-\pi^2 s_{10}(\pi^2 + s_2)^2(4\pi^2$

$+ s_2)\epsilon + e^{6\sqrt{s_2}}\pi^2 s_{11}(\pi^2 + s_2)(4\pi^2 + s_2)^2\epsilon - e^{\sqrt{s_2}}\pi^2 s_{14}(4\pi^2 + s_2)^2(4\pi^2 + 9s_2)\epsilon + e^{5\sqrt{s_2}}\pi^2 s_{15}(4\pi^2 +$

$s_2)^2(4\pi^2 + 9s_2)\epsilon + 12e^{2\sqrt{s_2}}\pi^2(\pi^2 + s_2)(4\pi^2 + 9s_2)(2A_1s_2(4\pi^2 + s_2) - (s_2(s_{13}(3 + \sqrt{s_2} + 2A_3s_2)$

$+ 4\pi^2(s_{13} + s_{13}\sqrt{s_2} + 2A_3s_2))\epsilon) - 12e^{4\sqrt{s_2}}\pi^2(\pi^2 + s_2)(4\pi^2 + 9s_2)(2A_2s_2(4\pi^2 + s_2) - (s_2(s_{12}(-3$

$+ \sqrt{s_2} + 2A_4s_2) + 4\pi^2(s_{12}(-1 + \sqrt{s_2}) + 2A_4s_2))\epsilon) + e^{-3\sqrt{s_2}}(-3s_2(\pi^2 + s_2)(4\pi^2 + s_2)(4\pi^2 + 9s_2)$

$(8A_1\pi^2 - 8A_2\pi^2 + \sqrt{s_2}(4\pi^2 + s_2)(2s_{20} + \alpha_c)) + pi^2(8\pi^4 s_2(3s_{10} - 3s_{11} + 96s_{12} + 96s_{13} + 22s_{14} - 22s_{15}$

$+ 168(A_3 - A_4)s_2) + \pi^2 s_2^2(9s_{10} - 9s_{11} + 900s_{12} + 900s_{13} + 76s_{14} - 76s_{15} + 1176(A_3 - A_4)s_2) + s_2^3(s_{10} -$

$s_{11} + 9(36s_{12} + 36s_{13} + s_{14} - s_{15} + 24(A_3 - A_4)s_2) + 16\pi^6(s_{10} - s_{11} + 4(3s_{12} + 3s_{13} + s_{14} - s_{15} +$

$6A_3s_2 - 6A_4s_2)))\epsilon)) \frac{e^{\alpha_c^2\tau - 3\sqrt{s_2} - 9\pi^2\tau}\cos(3\pi\eta)}{648s_2^{\frac{3}{2}}(s_2 + 9\pi^2)^2}(648A_1e^{2\sqrt{s_2}}(1 + e^{\sqrt{s_2}})s_2(9\pi^2 + s_2) - \frac{1}{\pi^2(\pi^2 + s_2)(9\pi^2 + 4s_2)}$

$(108e^{\sqrt{s_2}}\pi^4 s_{14}(\pi^2 + s_2)(9\pi^2 + s_2)^2\epsilon - 108e^{5\sqrt{s_2}}\pi^4 s_{15}(\pi^2 + s_2)(9\pi^2 + s_2)^2\epsilon + 3\pi^4 s_{10}(9\pi^2 + s_2)^2(9\pi^2$

$+ 4s_2)\epsilon - 3e^{6\sqrt{s_2}}\pi^4 s_{11}(9\pi^2 + s_2)^2(9\pi^2 + 4s_2)\epsilon + 324e^{2\sqrt{s_2}}\pi^4(\pi^2 + s_2)(9\pi^2 + 4s_2)(s_2(s_{13}(3 + \sqrt{s_2} +$

$2A_3s_2) + 2A_3s_2) + 9\pi^2(s_{13} + s_{13}\sqrt{s_2} + 2A_3s_2))\epsilon) + 324e^{4\sqrt{s_2}}\pi^4(\pi^2 + s_2)(9\pi^2 + 4s_2)(2A_2s_2(9\pi^2 + s_2) - (s_2(s_{12}(-3 + \sqrt{s_2} + 2A_4s_2) + 9\pi^2(s_{12}(-1 + \sqrt{s_2}) + 2A_4s_2))\epsilon) + e^{3\sqrt{s_2}}(648A_2\pi^4s_2(\pi^2 + s_2)(9\pi^2 + s_2)(9\pi^2 + 4s_2) + 4s_s^{\frac{3}{3}}(\pi^2 + s_2)(9\pi^2 + s_2)(9\pi^2 + 4s_2)(-4\alpha_c + 9\pi^2(4A_6 + 2s_{20} + \alpha_c)) + 3\pi^4(162\pi^4s_2(3s_{10} - 3s_{11} + 96s_{12} + 96s_{13} + 22s_{14} - 22s_{15} + 168(A_3 - A_4)s_2) + 9\pi^2s_2^2(9s_{10} - 9s_{11} + 900s_{12} + 900s_{13} + 76s_{14} - 76s_{15} + 1176(A_3 - A_4)s_2) + 4s_2^3(s_{10} - s_{11} + 9(36s_{12} + 36s_{13} + s_{14} - s_{15} + 24(A_3 - A_4)s_2) + 729\pi^6(s_{10} - s_{11} + 4(3s_{12} + 3s_{13} + s_{14} - s_{15} + 6A_3s_2 - 6A_4s_2)))\epsilon))$

Substituting $f_1$ in (38) and integrating this, we acquire numerical solutions of $K_2(\tau)$ with the support of the MATHEMATICA 8.0 software.

Similarly, $K_3(\tau)$, $K_4(\tau)$, etc., are considerably small in contrast to dispersion coefficient $K_2(\tau)$. Then, dispersion model Equation (24) becomes,

$$\frac{\partial \phi_m}{\partial \tau} = K_2 \frac{\partial^2 \phi_m}{\partial \xi^2} \tag{45}$$

with the help of Fourier Transform method (Rao [40]), we obtain the exact solution of (45), satisfying Equations (14)–(16),

$$\phi_m(\xi, \tau) = \frac{1}{2}\left[ erf\left(\frac{\frac{\xi_s}{2} + \xi}{2\sqrt{T}}\right) + erf\left(\frac{\frac{\xi_s}{2} - \xi}{2\sqrt{T}}\right)\right] \tag{46}$$

where, $T = \int\limits_0^\tau K_2(\eta)d\eta$ and $erf(\xi) = \frac{2}{\sqrt{\pi}}\int\limits_0^\xi e^{-z^2}dz$

## 4. Results and Discussion

With viscoelastic fluid confined by porous layers, the effects of electromagnetic fields and other physical parameters ($M = 0, 0.5, 1, 1.5$; $We = 10, 15, 20, 25$; $\epsilon = 1, 2, 3, 4$; $\sigma = (1, 5, 10)$) on cartilaginous cells are investigated. The perturbation technique and the GDM of Gill and SankaraSubramanian [28] were used to solve the nonlinear DE (9) and (10) numerically. To investigate the impact of the emerging parameters, computations were carried out with the parameters $Pe = 100, \alpha = 0.1$, and $Re = 0.05$ fixed. The obtained results were examined for different values of the relevant parameters.

Figures 2–5 elucidate the results of Equation (37), when used to calculate the dispersion coefficient with time, as well as the various impacts of the Hartmann and electric numbers, and the viscoelastic and porous parameters. According to Equation (37), $K_i's$ for $i > 2$ are insignificant in comparison to $K_2(\tau)$ and can be ignored. Equation (37) represents the unstable diffusion equation with a time-dependent $\hat{D}$. This figure shows that, for fixed values of $\alpha$, $Pe$, and $Re$, the dispersion coefficient increases as $\tau$ increases, and becomes free for large $\tau$. These figures demonstrate that the $K_k$ increases with certain parameters, such as $M$, $\epsilon$, $We$, and $\sigma$. The viscosity of the plasma of diabetics is greater than that of non-diabetics, often resulting in a higher diffusion coefficient for diabetic patients. The findings of this study are identical to the results of Alshehri and Sharma [1] on dispersing solutes.

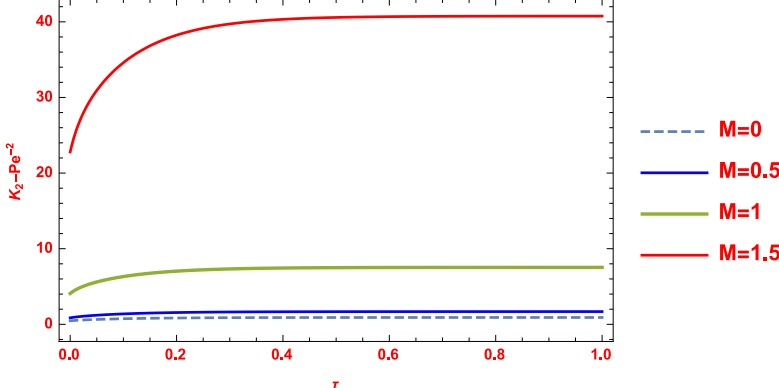

**Figure 2.** $K_2 - Pe^{-2}$ varying along $\tau$ for different $M$.

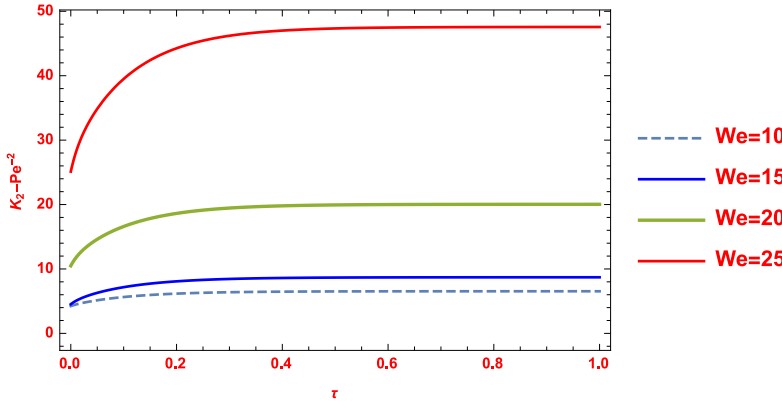

**Figure 3.** $K_2 - Pe^{-2}$ varying along $\tau$ for different $We$.

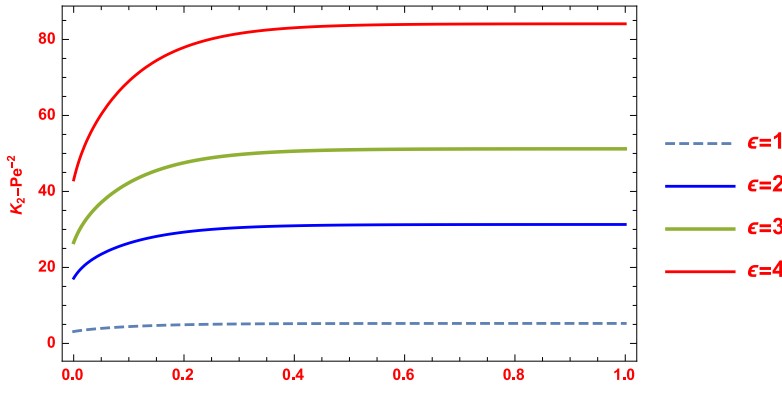

**Figure 4.** $K_2 - Pe^{-2}$ varying along $\tau$ for different $\epsilon$.

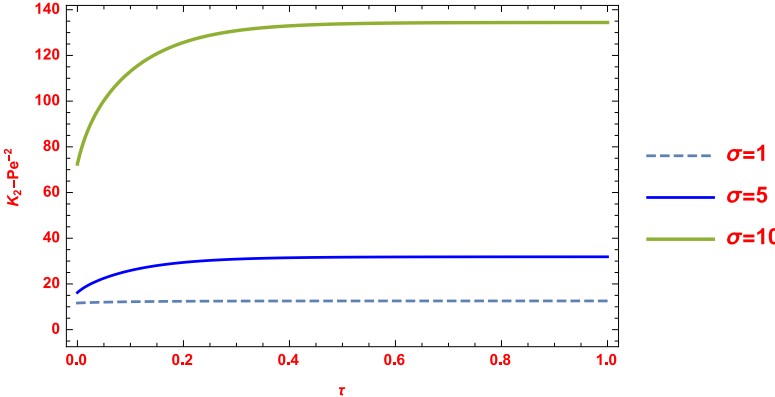

**Figure 5.** $K_2 - Pe^{-2}$ varying along $\tau$ for different $\sigma$.

The diffusion coefficient was calculated using the mean concentration from Equation (46). Figures 6–9 show the impacts of on the concentration profile varying along $\xi$ for a fixed time $\tau$. It is evident that the peak in mean concentration decreases as $M$, $\epsilon$, $We$, and $\sigma$ increase. It is symmetrical and bell-shaped at the origin. In the presence of Hartmann number and electric number, the Lorentz force, a resisting force created by the effects of a transverse magnetic field on an electrically conducting fluid, reduces the fluid's velocity and thickens the velocity and concentration boundary layer. The fluid concentration falls as a result of this. The presence of porous media decreases the velocity and concentration. We can see that the transfer of essential metabolites such as sugar and amino acids is quite sluggish, and convective transfer accelerates them from a medical perspective in biomechanics by means of mean concentration. As a result, the cells in the deep zone may receive fewer nutrients.

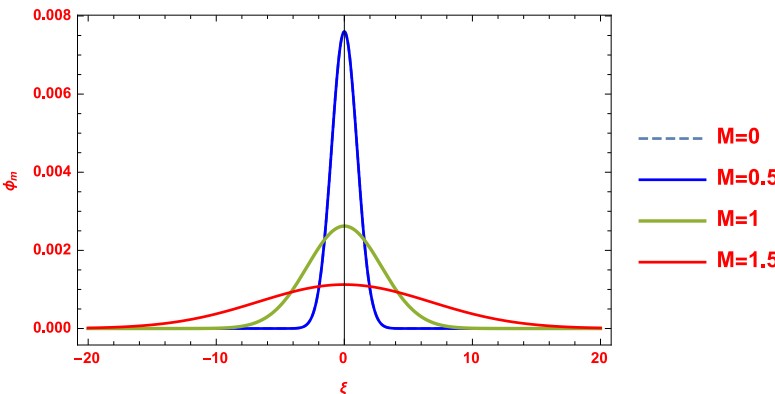

**Figure 6.** $\phi_m$ varying along $\xi$ for different $M$.

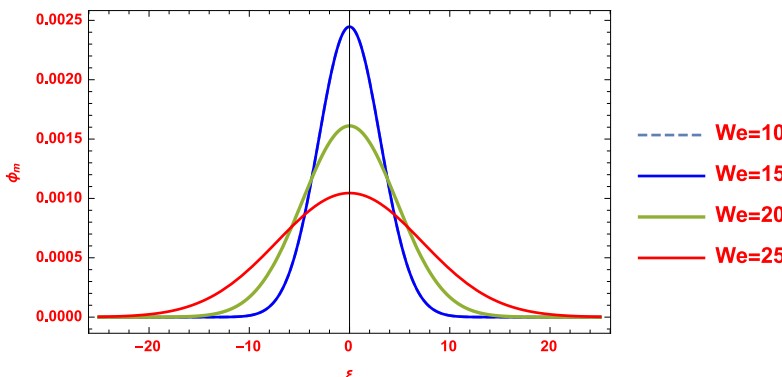

**Figure 7.** $\phi_m$ varying along $\xi$ for different $We$.

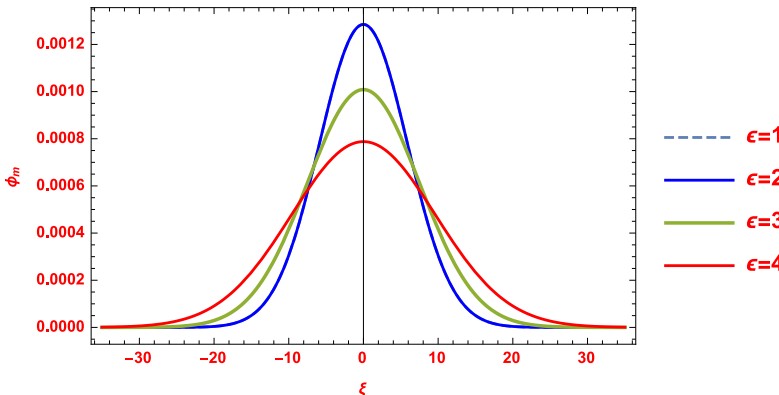

**Figure 8.** $\phi_m$ varying along $\xi$ for different $\epsilon$.

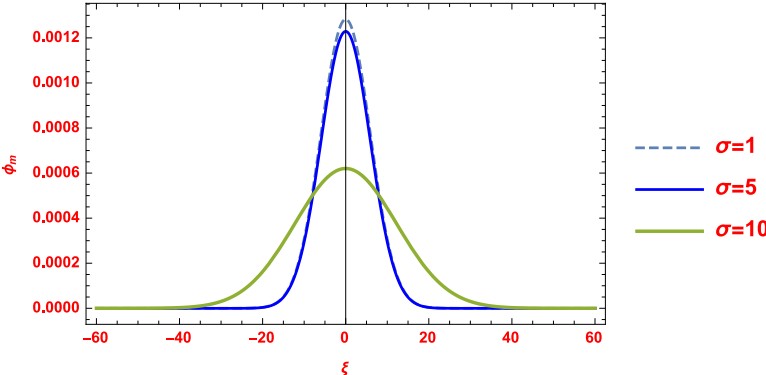

**Figure 9.** $\phi_m$ varying along $\xi$ for different $\sigma$.

Figures 10–13 show the impact of various factors ($M, We, \epsilon$, and $\sigma$) on the mean concentration $\phi_m$ in articular cartilage over time. The concentration decreases as one moves deeper into the articular surface. The figure also shows that concentration drops when $M$, $\epsilon$, $We$, and $\sigma$ are high. As time $\tau$ increases, the mean concentration of non-Newtonian fluid approaches zero. Analyzing solute transport over time is very useful. We can also see that fewer nutrients can flow through the cartilage pores. This is in good agreement with the results of Bali and Shukla [38].

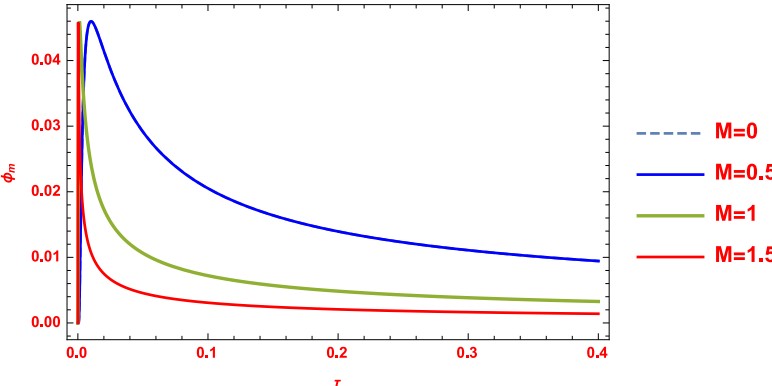

**Figure 10.** $\phi_m$ varying along $\tau$ for different $M$.

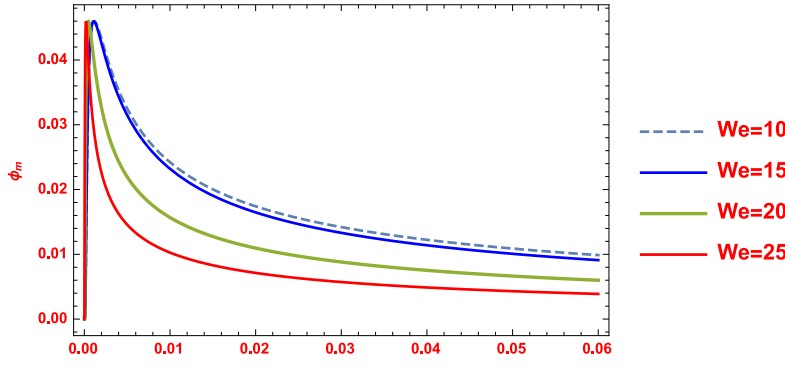

**Figure 11.** $\phi_m$ varying along $\tau$ for different $We$.

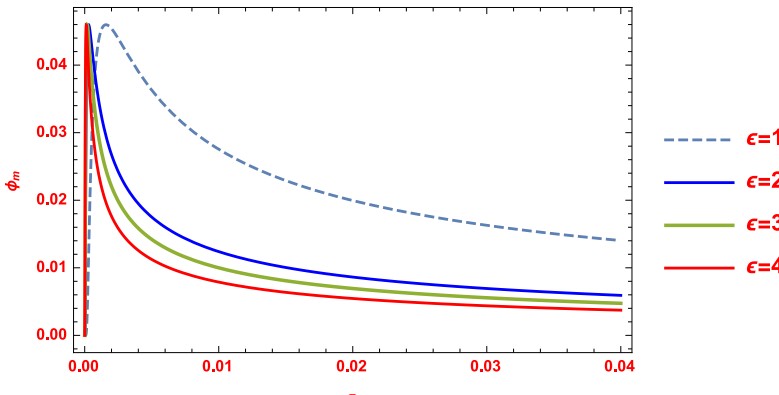

**Figure 12.** $\phi_m$ varying along $\tau$ for different $\epsilon$.

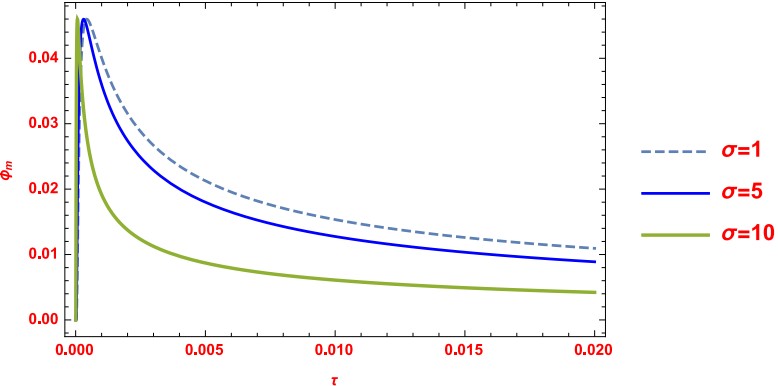

**Figure 13.** $\phi_m$ varying along $\tau$ for different $\sigma$.

## 5. Conclusions

The unsteady convective diffusion model was used to investigate the nutrient dispersion and other protein transport from the synovial fluid to articular cartilage using the perturbation method. This study's key conclusions are

- Dispersion is accelerated by electromagnetic fields and other physical factors.
- In contrast to electromagnetic fields and other physical factors, the mean concentration drops as axial distance and time increase.
- Cells in the centre receive more nutrients than those in the periphery.
- The dispersion mechanism formula is used by orthopaedic surgeons to assess how well joints function.

In the future, a mathematical model for articular cartilage regeneration could be created using the unsteady convective diffusion model, and this study could also be extended to investigate the spatial and temporal dynamics of nutrient diffusion and extracellular matrix depletion at the defect site.

**Author Contributions:** Conceptualization, B.R.K.; Writing—original draft, A.J.R.; Writing—review & editing, R.V. and B.R.K. All authors have read and agreed to the published version of the manuscript.

**Funding:** This research received no external funding.

**Conflicts of Interest:** The authors declare no conflict of interest.

**Nomenclature**

| | |
|---|---|
| $(\hat{u}, \hat{v})$ | Horizontal and normal components of the fluid velocity |
| $(\hat{x}, \hat{y})$ | Cartesian coordinates |
| $\hat{p}$ | Pressure |
| $\tilde{u}$ | average velocity |
| $B_0$ | Magnetic induction |
| $k$ | Permeability of porous medium |
| $\hat{E}_x$ | x component of electric field |
| $\hat{C}$ | Species concentration |
| $\hat{C}_0$ | Initial species concentration |
| $\hat{D}$ | Diffusion coefficient |
| $K_k$ | Dispersion coefficient |
| $M$ | Hartmann number |
| $We$ | Electric number |
| $Re$ | Reynolds number |
| $Pe$ | Peclet number |
| $Da$ | Darcy number |

| | |
|---|---|
| *Greek Symbols* | |
| $\mu$ | Dynamic viscosity |
| $\hat{\eta}$ | Kinematic viscosity |
| $\alpha$ | Slip parameter |
| $\sigma_0$ | Electrical conductivity |
| $\sigma$ | Porous parameter |
| $\epsilon$ | Viscoelastic parameter |
| $\phi$ | Concentration |
| $\phi_m$ | Mean concentration |
| $\rho_e$ | Dimensionless charge density |
| $\tau$ | Dimensionless time |
| $\bar{\xi}$ | Dimensionless axial distance |

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
