# Peer review of "A Model for the Generalised Dispersion of Synovial Fluids on Nutritional Transport with Joint Impacts of Electric and Magnetic Field"

_mca, doi:10.3390/mca28010003_

Round 1

Reviewer 1 Report

Dear authors,

Your manuscript's topic is interesting but there are some weaknesses of the manuscript that need to be addressed for correction as follows:

1. The proposed governing equations (1)-(3) are not from the original PDE basis and that they are the reduced form of which the authors did not exposed what assumptions have been taken prior to the reduction/simplication process of the equations. No reliable reference has been cited to support the equations either. Note that the base PDE model must be non-Newtonian.

2. The physical configuration in Figure 1 showed the magnetic field is imposed from outside the cell surface eg. the skin. How such visualization makes sense in the practical synovial fluid related treatment? Justify this in the revised manuscript sufficiently.

3. In separate pages, prove how Equations (1)-(3) are derived and how Equations (9)-(10) are derived clearly for the reviewer's validation. There are some defined scaling parameters which are not consistent with the terms in Equations (9)-(10). Correct them sufficiently.

4. From Equation (20) onwards, theta was used to denote concentration instead of heat whereas in the earlier part of the manuscript, authors claimed to solve momentum and heat equations only.

5. The graphical results from Figure 2 onwards were produced without any validation done to confirm the reliability and correctness of the proposed solutions. The discussions for these figures are also shallow without sufficient physical interpretation.

6. Authors should present the corresponding table of  numerical results for future comparison and validation by other researchers to extend this field.

7. How diabetic issue is related to the figures' discussion using the predefined parameters? Where is the term referring to diabetic from the original equations of the problem?

8. Explain clearly how/why concentration drops with increments in magnetic, viscoelastic, porosity and electric parameters?

9. Highlight the velty of the present work as compared to the previous works by other authors in Abstract, last paragraph of introduction and in the conclusion section.

10. The following works are recommended to enrich the literatures on non-Newtonian flows for bio applications:

a) Current Nanoscience 10, 432-440, 2014

b) Case Studies in Thermal Engineering 35, 102124, 2022

Author Response

Reviewer-1

Q1.The proposed governing equations (1)-(3) are not from the original PDE basis and that they are the reduced form of which the authors did not exposed what assumptions have been taken prior to the reduction/simplification process of the equations. No reliable reference has been cited to support the equations either. Note that the base PDE model must be non-Newtonian.

Author Response: Clear idea of governing equations and Reliable reference included in the revised manuscript.

 Tandon, P.N.; Nirmala, P.; Pal, T.S.; Agarwal, R. Rheological study of lubricant gelling in synovial joints during articulation. Applied Mathematical Modelling 1988, 12, 72–77.

 Bali, R., Shukla, A. Rheological effects of synovial fluid on nutritional transport. Tribology Letters 9, 233–239 (2001)

Formulation of the problem

Knee joint is called patellofemoral joint which is between (femur) the end of the thigh bone and (patella) the kneecap. It plays vital role in the human body for everyday activities. Figure 1 constitutes knee joint and the determination of systematic treatment; the knee joint composed of fibrous, cartilaginous and synovial. The geography of carriage model be composed of two equal – sided plates of unbounded length in x direction than the width in y direction. Cartilaginous is symbolized on the thickness of porous-viscoelastic pad. The equal – sided surfaces proceed towards each other with a squashing velocity with respect to some gap 2h.

With the purpose of framing a mathematical formulation for these problems, we consider the following assumption.  

  • Depending on the structure, Cartilage in human body can be segregated as elastic.
  • Magnetic effect through porous medium is considered.
  • The solid and liquid phases, which are isotropic, homogenous, and incompressible.
  • Solidness to fluidness ratio (v) is constant.
  • Inertial forces are insignificant during articulation due to tiny transients.

The governing differential equations of continuity, momentum, and constitutive relations for each phase of the cartilage matrix and the fluid film region are provided separately below under these presumptions.

Q2. The physical configuration in Figure 1 showed the magnetic field is imposed from outside the cell surface eg. the skin. How such visualization makes sense in the practical synovial fluid related treatment? Justify this in the revised manuscript sufficiently.

Author Response: The externally applied magnetic field has, therefore, considerable effects on biological systems. It is believed that the applied magnetic field stimulates the functions of the various physiological systems and also regenerates the tissues in the body. An applied magnetic field, accelerates bone in-growth into the implants and simultaneously increases the load carrying capacity with the reduction of friction. It has been established that human joints, in general, are greatly affected by the application of external magnetic fields. The applied magnetic fields may be used for better articulation, particularly in diseased states.

Q3. In separate pages, prove how Equations (1)-(3) are derived and how Equations (9)-(10) are derived clearly for the reviewer's validation. There are some defined scaling parameters which are not consistent with the terms in Equations (9)-(10). Correct them sufficiently.

Author Response: Derivation of equations (1) – (3) and (9) – (10) are given in the following reference papers. The flow of incompressible viscoelastic fluids for which inertia is neglected is governed by the momentum and mass balance equations, ∇ p − ∇ · ( 2 η s D ) − ∇ · τ = 0 and ∇ · u = 0

Tandon, P.N.; Nirmala, P.; Pal, T.S.; Agarwal, R. Rheological study of lubricant gelling in synovial joints during articulation. Applied Mathematical Modelling 1988, 12, 72–77.

 Bali, R., Shukla, A. Rheological effects of synovial fluid on nutritional transport. Tribology Letters 9, 233–239 (2001)

Q4. From Equation (20) onwards, phi was used to denote concentration instead of heat whereas in the earlier part of the manuscript, authors claimed to solve momentum and heat equations only.

Author Response: In the revised manuscript, the theta is changed into phi. We have considered and solved momentum and concentration equations.

Q5. The graphical results from Figure 2 onwards were produced without any validation done to confirm the reliability and correctness of the proposed solutions. The discussions for these figures are also shallow without sufficient physical interpretation.

Author Response: In the revised version, discussion part has been enhanced and added adequate significance of the physical interpretation. The Physical description of the graphs also included in the revised version.

Q6. Authors should present the corresponding table of numerical results for future comparison and validation by other researchers to extend this field.

Author Response: We have validated our work with the results of Bali and Shukla [25] through Figures 10-13. Also, we see that fewer nutrients can flow through the cartilage pores. This is in good agreement with the results of Bali and Shukla [25].

Q7. How diabetic issue is related to the figures' discussion using the predefined parameters? Where is the term referring to diabetic from the original equations of the problem?

Author Response: The diffusion coefficient is calculated using the mean concentration from equation 46. We notice that the transfer of essential metabolites like sugar and amino acids is quite sluggish, and convective transfer accelerates them from a medical point of view in biomechanics by means of mean concentration. As a result, the cells in the deep zone may get fewer nutrients. The viscosity of the plasma of diabetic is greater than that of non-diabetic, which results in a higher diffusion coefficient for diabetic patients.

Q8. Explain clearly how/why concentration drops with increments in magnetic, viscoelastic, porosity and electric parameters?

Author Response: The effects of magnetic, viscoelastic, porosity and electric parameters accelerate the fluid particles in the flow domain. Viscoelastic donates a sudden fall and magnetic force donates to greater skin friction as well as porosity and electric parameters happen to be the same. Hence Destructive reaction reduces whereas generative reaction enhances the concentration distribution. 

Q9. Highlight the novelty of the present work as compared to the previous works by other authors in Abstract, last paragraph of introduction and in the conclusion section.

Author Response: In the revised version, novelty of the present work is included in the abstract, the last paragraph of introduction and conclusion sections and the same is highlighted in the revised version.

The important contribution of this article is to create a model for synovial fluid which is known as joint fluid located in the knee joints examining the nutritional transportation of generalized dispersion with the effect of electric and magnetic field. The behavior of the synovial fluid is criticized by the perturbation technique and generalized dispersion model. The exact solution is plotted graphically and explained and discussed in detail.

Q10. The following works are recommended to enrich the literatures on non-Newtonian flows for bio applications:

  1. a) Current Nanoscience 10, 432-440, 2014
  2. b) Case Studies in Thermal Engineering 35, 102124, 2022

Author Response: The suggested papers are included in introduction and References.

Reviewer 2 Report

The authors have studied the "A Model for the Generalised Dispersion of Synovial Fluids on Nutritional Transport with Joint Impacts of Electric and Magnetic Field". The topic is very interesting and therefore, I recommend the paper for a possible publication in the given journal. Before acceptance, the authors need to address the following minor comments.

1. Please modify the novelty part of the given paper. It is better to full fill the research gap.

2. Improve the physical model of the problem. Also, written everything clear in the picture that the readers can easily understand.

3. Please rewrite the main assumption of the problems in bullets form.

4. Why the convective term goes to zero in momentum equation. Please write in full detail.

5. Explain the physical significance of the boundary conditions.

6. Improve results and discussion section by adding more physical interpretations.

7. Add future direction of the given model at the end of the conclusion.

8. Please reduce the similarity of the paper less than 20. Also, try to reduce the individual similarity less than 1 or equal to 1.

9. To improve the introduction section then try to cite the following papers.

a. Khan, U., Zaib, A., Shah, Z., Baleanu, D. and Sherif, E.S.M., 2020. Impact of magnetic field on boundary-layer flow of Sisko liquid comprising nanomaterials migration through radially shrinking/stretching surface with zero mass flux. Journal of Materials Research and Technology9(3), pp.3699-3709.

b. Khan, U., Zaib, A., Khan, I. and Nisar, K.S., 2021. Insight into the dynamics of transient blood conveying gold nanoparticles when entropy generation and Lorentz force are significant. International Communications in Heat and Mass Transfer127, p.105415.

Author Response

Reviewer-2

Q1. Please modify the novelty part of the given paper. It is better to full fill the research gap.

Author Response: The novelty part of the paper is revised in the current manuscript.

The important contribution of this article is to create a model for synovial fluid which is known as joint fluid located in the knee joints examining the nutritional transportation of generalized dispersion with the effect of electric and magnetic field. The behavior of the synovial fluid is criticized by the perturbation technique and generalized dispersion model. The exact solution is plotted graphically and explained and discussed in detail.

Q2. Improve the physical model of the problem. Also, written everything clear in the picture that the readers can easily understand.

Author Response: The physical model of the problem is improved in the revised version

Q3. Please rewrite the main assumption of the problems in bullets form.

Author Response: This is done in the revised version.

Q4. Why the convective term goes to zero in momentum equation. Please write in full detail.

Author Response: The terms on the left hand side of the momentum equations are called the convection terms of the equations. Due to the non-motion of the fluid the convective term become zero.

Q5. Explain the physical significance of the boundary conditions.

Author Response: In the revised version, the physical significance of the boundary conditions are explained.

Q6. Improve results and discussion section by adding more physical interpretations.

Author Response: It is improved in the revised manuscript.

Q7. Add future direction of the given model at the end of the conclusion.

Author Response: Future scope of the problem is added at the end of the conclusion

Q8. Please reduce the similarity of the paper less than 20. Also, try to reduce the individual similarity less than 1 or equal to 1.

Author Response: It is reduced in the current version.

Q9. To improve the introduction section then try to cite the following papers.

  1. Khan, U., Zaib, A., Shah, Z., Baleanu, D. and Sherif, E.S.M., 2020. Impact of magnetic field on boundary-layer flow of Sisko liquid comprising nanomaterials migration through radially shrinking/stretching surface with zero mass flux. Journal of Materials Research and Technology9(3), pp.3699-3709.
  2. Khan, U., Zaib, A., Khan, I. and Nisar, K.S., 2021. Insight into the dynamics of transient blood conveying gold nanoparticles when entropy generation and Lorentz force are significant. International Communications in Heat and Mass Transfer127, p.105415.

Author Response: The suggested papers are cited in the revised version

Reviewer 3 Report

Minor comments:

Comment No. 1: This paper should be edited grammatically.

Comment No. 2: You should add quantitative results in the abstract.

Comment No. 3: You should double check the mathematical formulations.

Comment No. 4: The originality of the paper needs to be stated clearly. It is of importance to have sufficient results to justify the novelty of a high-quality journal paper. The Introduction should make a compelling case for why the study is useful along with a clear statement of its novelty or originality by providing relevant information and providing answers to basic questions such as: What is already known in the open literature? What is missing (i.e., research gaps)? What needs to be done, why and how? Clear statements of the novelty of the work should also appear briefly in the Abstract and Conclusions sections.

Comment No. 5: An updated and complete literature review should be conducted and should appear as part of the Introduction, while bearing in mind the work's relevance to this Journal and taking into account the scope and readership of the journal. The results and findings should be compared to and discussed in the context of earlier work in the literature.

Comment No. 6: You should add a nomenclature.

Comment No. 7: The literature review is inadequate; you should include relevant publications such as:

10.1016/j.padiff.2022.100450, 10.1016/j.taml.2022.100360, 10.1007/s13204-020-01483-y,

Major comments:

Comment No. 1: How the similarity parameters have been obtained?

Comment No. 2: Result and discussion section can be more improved from the physical point of view.

Comment No. 3: what is your method advantage?

Comment No. 4: You should provide more information about your validation. 

Author Response

Reviewer-3

Minor comments:

Comment No. 1: This paper should be edited grammatically.

Author Response: As per suggestion the revised paper is edited grammatically

Comment No. 2: You should add quantitative results in the abstract.

Author Response: As per suggestion, results are added in the abstract

Comment No. 3: You should double check the mathematical formulations.

Author Response: Verified the mathematical Formulation and corrected in the revised version.

Comment No. 4: The originality of the paper needs to be stated clearly. It is of importance to have sufficient results to justify the novelty of a high-quality journal paper. The Introduction should make a compelling case for why the study is useful along with a clear statement of its novelty or originality by providing relevant information and providing answers to basic questions such as: What is already known in the open literature? What is missing (i.e., research gaps)? What needs to be done, why and how? Clear statements of the novelty of the work should also appear briefly in the Abstract and Conclusions sections.

Author Response: In the revised manuscript, the novelty of the manuscript is added in abstract, introduction and in the conclusions section

Comment No. 5: An updated and complete literature review should be conducted and should appear as part of the Introduction, while bearing in mind the work's relevance to this Journal and taking into account the scope and readership of the journal. The results and findings should be compared to and discussed in the context of earlier work in the literature.

Author Response: The literatures relevant to our research problem are cited properly from this journal

Comment No. 6: You should add a nomenclature.

Author Response: The nomenclature has been added in the revised version

Comment No. 7: The literature review is inadequate; you should include relevant publications such as:

10.1016/j.padiff.2022.100450, 10.1016/j.taml.2022.100360, 10.1007/s13204-020-01483-y,

Author Response: Related literatures review has been addedin the revised version of the manuscript.

Major comments:

Comment No. 1: How the similarity parameters have been obtained?

Author Response: similarity is the basis for simulating large-scale phenomena in a small-scale laboratory. The approach of making approximations based on scaling and normalization.

Comment No. 2: Result and discussion section can be more improved from the physical point of view.

Author Response: In the revised version, we have updated the results and discussion section with physical description

Comment No. 3: what is your method advantage?

Author Response: Perturbation techniques are a class of analytical methods for determining approximate solutions of nonlinear equations for which exact solutions cannot be obtained. They are useful for demonstrating, predicting, and describing phenomena in vibrating systems that are caused by nonlinear effects.

Comment No. 4: You should provide more information about your validation.

Author Response: From Figures 10-13, we have shown the  impact of various factors (M, We,   , and ) on concentration with time on the mean concentration  in articular cartilage. The concentration decreases as one move deeper into the articular surface. The figure also shows that concentration drops when M, ,  We, and  are high. As time  increases, the mean concentration of non-Newtonian fluid approaches zero. Analyzing solute transport over time is very useful. Also, we see that fewer nutrients can flow through the cartilage pores. This is in good agreement with the results of Bali and Shukla [25]. Hence our results are Valid.

Round 2

Reviewer 3 Report

Now it can be published.

Author Response

In the revised manuscript, the English standard has been improved by proofreading and editing. Also, We verified spelling check and corrected throughout the manuscript